# Polycystic Ovary Syndrome in Adolescence

**DOI:** 10.3390/medsci7100101

**Published:** 2019-10-02

**Authors:** Rebecca Deans

**Affiliations:** University of New South Wales, Royal Hospital for Women, Sydney Children’s Hospital, Sydney 2031, Australia; r.deans@unsw.edu.au

**Keywords:** polycystic ovary syndrome, adolescence

## Abstract

Polycystic ovary syndrome (PCOS) is one of the most common endocrine disorders in females, and is characterized by ovulatory dysfunction, hyperandrogenism, and polycystic ovarian morphology based on ultrasound. Controversy surrounds the optimum diagnosis and management in the adolescent population. Many patients with adult PCOS present with pathognomonic symptoms as adolescents, and there is value in early diagnosis due to the associated long-term metabolic and reproductive health sequalae. A definitive diagnosis does not need to be made prior to implementing treatment in this group of young women. The practitioner who has an adolescent presenting with signs and symptoms of PCOS, has a unique opportunity to risk stratify, screen for co-morbidities, and implement early management strategies, many of which are lifestyle modifications, to help prevent long term morbidity associated with this disease.

## 1. Introduction

Polycystic ovary syndrome (PCOS) is one of the most common endocrine disorders in females. The reported prevalence of PCOS in the community is between 6–10% depending on which criteria is used to define it [1]. PCOS is characterized by ovulatory dysfunction, hyperandrogenism, and polycystic ovarian morphology (PCOM) based on ultrasounds [2]. Despite the high prevalence of this condition, controversy surrounds the optimum diagnostic criteria and management for the adolescent population. Hyperandrogenism is the most consistent characteristic of PCOS in both adults and adolescents [3,4]. Adult patients with PCOS commonly present with pathognomonic symptoms during adolescence. There is value in early identification of PCOS to optimally manage the associated long-term metabolic and reproductive health sequalae [5]. Treatment should be tailored to the individual and account for the nuances of this chronic condition when diagnosed in young women.

## 2. Pathogenesis and Clinical Features

Clinical features of PCOS include clinical hyperandrogenism in the form of hirsutism, acne, or alopecia. Menstrual irregularity encompasses primary or secondary amenorrhea, oligomenorrhea, irregular periods, and heavy menstrual bleeding. Clinical tests show polycystic ovarian morphology (PCOM) on ultrasound, and / or metabolic derangement on blood testing, including insulin resistance, glucose intolerance, obesity and dyslipidaemia. There can be a marked heterogenicity in its clinical presentation [6].

There is a complex relationship with genetic, metabolic, endocrine, environmental, and lifestyle factors in PCOS, and the aetiology that remains poorly understood [7,8,9,10]. Established theories include disordered neuroendocrine gonadotropin secretion, hyperandrogenism, insulin resistance, or a combination of these [11].

## 3. Diagnostic Criteria

The Rotterdam criteria [12], is the most commonly accepted criteria for the diagnosis of PCOS in adults [2]. The definition of polycystic ovarian morphology (PCOM) was 12 or more follicles on the ovary measuring 2–9 mm, or an ovarian volume of at least 10 cm^3^. It is now understood that 70% of the adolescent population will have PCOM based on these criteria [13]. Clinical hyperandrogenism in the form of acne vulgaris (AV) is also a common feature of adolescence. Therefore, the adult diagnostic criteria for PCOS may be unreliable when applied to AYAs, as the diagnostic features often overlap with normal physiological changes of puberty, including irregular menstrual cycles and acne [14,15,16,17,18,19,20,21,22]. Several diagnostic criteria for PCOS have been specifically proposed for AYAs, including: The 2012 Embryology/American Society (EAS) for Reproductive Medicine Consensus workshop group reported that all three of the Rotterdam criteria should be fulfilled for the diagnosis of PCOS in adolescents [15,23,24]. However, there may be adolescents who do not meet all three criteria that still manifest signs of hyperandrogenism or irregular menses. Hence, these patients may be at risk of developing PCOS in the future, and it is recommended that these individuals to be reviewed at regular intervals (annually or biennially) to assess the possible onset of PCOS [15].In 2015, a consensus panel convened by the Pediatric Endocrine Society (PES) developed another set of diagnostic criteria for adolescents [9]. The PES criteria require the presence of unexplained persistent hyperandrogenic and oligo-anovulatory menstrual abnormalities, based on age and stage appropriate standards.In 2018, international evidence-based guidelines stated that for young patients <8 years post menarche, both hyperandrogenism and ovulatory dysfunction must be present for the diagnosis of PCOS to be made [2]. The appearance of polycystic ovarian morphology (PCOM) is of less value due to its lack of reliability as a pathologic finding in the 8 years following menarche. There was also a revision in the ultrasound measurement of the follicle number per ovary (FNPO), with a recommendation of a FNPO of at least 20 and/or ovarian volume of at least 10 mL to ensure no dominant follicles are present [2].

Differentiating between normal immature hypothalamic pituitary ovarian (HPO) axis caused physiological anovulation and true ovulatory dysfunction is problematic in adolescents [9,23,25,26]. However, it is generally agreed that adolescent menstrual cycles fall within certain parameters, and therefore evidence of ovulatory dysfunction includes: (I) consecutive menstrual intervals >90 days apart for over a year following menarche, (II) menstrual intervals persistently <21 or >45 days, two or more years following menarche, and (III) lack of menses by the age of 15, or two to three years after breast budding [25]. 

Although isolated mild hyperandrogenism is considered normal in early post-menarchal years, moderate to severe hirsutism is clinical evidence of androgen excess [9], as is persistent acne (longer than six months) which is unresponsive to topical treatment. Studies of adolescent girls with PCOS have shown that free androgen index (FAI) is significantly higher and sex hormone binding globulin (SHBG) is significantly lower in overweight and obese patients, compared to those with low body mass index (BMI) [27,28]. Obesity can cause functional hyperandrogenism due to reduced SHBG synthesis [29].

When assessing hyperandrogenism, it is important for clinicians to exclude the presence of other pathologies, particularly those that may manifest in oligo-amenorrhea and hyperandrogenism. These include thyroid dysfunction, prolactinemia, hypercortisolemia, congenital adrenal hyperplasia (CAH), hyperthecosis, exogenous androgen exposure, and androgen-producing tumours. Clinical and biochemical investigations should differentiate these conditions.

## 4. Investigations

Measurement of total and free testosterone remains the mainstay of the biochemical analysis of hyperandrogenemia. However, there is a lack of consensus in regard to the preferred androgen assay and reference values for adolescents [5,18,23,30]. This is due to biological variables that affect testosterone levels, such as the diurnal rhythm, stage of puberty, phase of menstrual cycle and sex hormone binding globulin levels [20]. Elevated circulating androgen levels are observed in 60–80% of PCOS patients [31]. As a single measure, calculated free testosterone, free androgen index (FAI), and calculated bioavailable testosterone are the most accurate tests in the assessment of biochemical hyperandrogenemia in PCOS [32]. In most laboratories, the upper limit of 55 ng/dL [9,26] for total testosterone and 9 pg/mL for free testosterone is used [9]. The most reliable assays include liquid chromatography–mass spectrometry (LCMS), mass spectrometry and extraction, and chromatography immunoassays [2]. Hormone levels should be preferably drawn in the morning, and it is important to remember that other treatments, such as the combined oral contraceptive pill (COCP), will affect the laboratory androgen results.

Laboratory workup should be individualised to exclude other causes of hyperandrogenism. Generally, this workup includes 17-hydroxyprogesterone (17-OHP), androstenedione, free thyroxine (FT4), thyroid stimulating hormone (TSH), luteinising hormone (LH), follicle stimulating hormone (FSH), and prolactin. Dehydroepiandrosterone sulphate (DHEAS) and androstenedione have a limited role in assessment of PCOS, but are useful in the exclusion of other causes of hyperandrogenemia, including CAH, although this may be mildly elevated in AYAs with PCOS. If there is an abnormality in androstenedione or 17-OHP, a cosyntropin (ACTH) stimulation test should be ideally performed to screen for non-classical congenital adrenal hyperplasia (CAH) [33]. Pregnancy should be excluded in patients who are sexually active.

A LH/FSH ratio >2 has been used as part of diagnostic criteria for PCOS diagnosis in adolescents [34], but controversy remains with regards to its value. Some authors report significantly higher LH levels and LH/FSH ratios in PCOS patients [35], but this is not replicated in other studies [36]. Using a LH/FSH ratio >2 as a criterion may be misleading in AYA patients with PCOS.

More recently, anti-Müllerian hormone (AMH) has been studied in AYAs as a predictor of adult PCOS. Correlations were made between elevated AMH levels at age 16, and higher measures of testosterone, clinical hirsutism, menstrual irregularity, as well as diagnosis of PCOS by the age of 26 [37,38]. Although AMH shows promise as an adjunct to the diagnostic tests recommended for PCOS, its utilization in adolescence needs to be considered carefully as there is a wider range of normal values in AMH seen in AYAs in population-based studies [39].

Pelvic ultrasonography is generally not recommended for the diagnosis of PCOS in adolescence. The ultrasound criteria for the diagnosis of PCOS are not well defined. However, pelvic ultrasound may be used to exclude other underlying pathologies, based on clinical features.

## 5. Co-Morbidity

Early assessment of the co-morbidities associated with PCOS is an important component of the management of the AYA, as it affords the benefit of time to manage and treat these conditions as they arise [5]. Clinically significant metabolic and psychological sequalae have described PCOS in adolescence as an independent risk factor for disease [37,38]. Insulin resistance and hyperinsulinism have also been well documented in PCOS [26], as is hyperlipaemia, and obesity confers its own risk. [29]. Therefore, patients should be screened for the presence of co-morbidities as part of routine clinical care.

Insulin resistance is ideally identified by euglycemic clamp, but as this test is laborious and invasive, fasting insulin is often used as an alternative. However, this test lacks sensitivity as hyperinsulinemia is a feature of normal puberty [39]. The 2 h plasma glucose tolerance test (GTT) is the most reliable screening test, and some advocate interval screening of adolescents with PCOS [40].

The co-existence of psychological disorders with PCOS has been documented and assessment of these symptoms should be considered in AYAs with PCOS [37]. Depression and anxiety symptoms are more common in PCOS women compared with BMI matched subjects [41,42] although these data are not focused on an adolescent population. It is uncertain whether these changes relate to clinical symptoms such as acne, hirsutism, increased BMI, and infertility [2], or whether it is associated to the chronic nature of PCOS [43]. In addition, negative body image and eating disorders are more prevalent in adolescent PCOS, and clinicians should be cognisant of this when giving advice to AYAs regarding lifestyle and weight loss [44].

## 6. Management

Treatment should be tailored to the needs of the individual. The aims of treatment are to improve quality of life and long term health outcomes, whilst balancing the side effects of treatment [45]. 

According to the consensus document [26], additional management considerations for adolescents with PCOS symptoms include: (i)A definitive diagnosis of PCOS is not necessary prior to initiating treatment. Treatment may decrease risk of future co-morbidity, even in the absence of a definitive diagnosis.(ii)Deferring the diagnosis of PCOS while offering symptomatic treatment and providing regular follow up is recommended.(iii)Obesity, hyperinsulinemia, and insulin resistance are recognised as common in adolescents with PCOS.(iv)Other causes of hyperandrogenemia and irregular menstrual periods must be ruled out before a diagnosis of PCOS can be established.

Lifestyle modifications remain the first line treatment in AYAs who are overweight or obese. Improvement in cycle regularity, reduction in metabolic and cardiac risk factors, and reduction in hyperandrogenism have been demonstrated in this population with weight loss alone [46,47].

Cosmetic hair removal offers a more immediate result for girls with hirsutism and may be required in conjunction with medical management. Waxing, electrolysis, and laser hair removal are all commonly utilised, with the latter becoming more popular and affordable. Topical hair removal, including eflornithine cream, requires continuous use. Combining this cream with laser therapy has been shown to reduce hirsutism [48,49]. Side effects of eflornithine cream include local skin irritation, pseudofolliculitis barbae, headache, nausea, and occasionally urticaria and anaphylaxis. 

Combined oral contraceptive pills (COCP), are commonly used as first line medical management for AYAs with irregular menstrual cycles, and also improves acne and hirsutism [50]. An additional benefit is contraception, which may be necessary for young women who are sexually active. The pill reduces ovarian androgen production by suppressing the HPO axis and in addition, the increase in sex hormone binding globulin (SHBG) decreases free androgen [51]. The progestogen component of the COCP also prevents unopposed estrogen action and subsequent endometrial hyperplasia.

Metformin is recommended for women with type II diabetes or impaired glucose tolerance which is unresponsive to lifestyle modifications [50]. Metformin has been shown to be as effective in the management of hirsutism as the COCP, and was superior to the COCP for controlling weight related symptoms and glucose and insulin dysregulation, but the COCP was superior for menstrual regulation [45]. Therefore, metformin should be considered for AYAs with PCOS symptoms including hirsutism, metabolic derangement, and glucose intolerance. For symptoms of menstrual irregularity, it is likely to be inferior to the COCP, but may improve symptoms compared to no treatment.

The anti-androgenic medication, oral spironolactone, has been used as an adjunct to the COCP and Metformin. This may improve menstrual regulation as well as clinical cutaneous androgenic symptoms, but has no effect on metabolic dysregulation [52,53]. Spironolactone can be initiated as monotherapy in AYAs with androgenic symptoms such as AV, or it can be used in combination with other topical and oral agents. The efficacy of spironolactone therapy has been reported in several studies, with lesion reductions ranging from 50–100% in women treated for AV [54]. Caution should be exercised with AYAs with renal disease as spironolactone may cause hyperkalaemia, and as there are teratogenic effects, contraception must be used in conjunction with this medication. Finasteride has also been shown to be an effective treatment for adolescent girls with hirsutism as monotherapy or in combination with the COCP [55,56]. Finally, access to dieticians and psychologists is an important component to the management of AYAs with PCOS due to the high psychological burden of this disease. Improving the psychological support and focusing on lifestyle modification is the key to managing this condition in its early stages. 

Tools for the assessment of the psychological distress in AYAs include: Screening for Depression in Children and Adolescents: US Preventative Services Task Force Recommendations [57].Common mental health problems: Identification and pathways to care, NICE, 2011 [58].Screening for and Treatment of Suicide Risk Relevant to Primary Care: A Systematic Review for the US Preventative Services Task Force [59].

## 7. Conclusions

Many adults diagnosed with PCOS present with symptoms in their AYA years. Consideration of this condition and early investigation are important for instituting management and prevention strategies to minimise the long-term heath sequelae of PCOS. Screening for the psychological burden is recommended. The diagnosis of PCOS in adolescents is problematic, as criteria are less reliable than in the adult population. It is not necessary to make a definitive diagnosis prior to the commencement of treatment. Early identification of the co-morbidities and targeted management focusing on lifestyle modification is the key to mediating risk in AYAs with this chronic disease.

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
