# Peer review of "Polycystic Ovary Syndrome in Adolescence"

_medsci, 2019, doi:10.3390/medsci7100101_

Round 1

Reviewer 1 Report

This brief review of PCOS in adolescence addresses most of the issues regarding presentation, evaluation, and management of the disorder. It is unclear whether the brevity of the piece is due to space limitation or the choice of the author. Specific comments are as follows:

LL 21-22: It would seem as if the discussion of criteria should be expanded to include, at least, Rotterdam criteria. This would raise the prevalence of PCOS to at least 20-24%. Also, Rotterdam criteria are mentioned later in the text.  

LL33-35: This does not seem to be a complete sentence.

LL 67: “…hyperandrogenism…” is misspelled. Also, should the verb be “is” instead of “was”?

LL 76-78: Would the author include CAH, hyperthecosis, and exogenous androgen exposure?

LL 81-82: It would appear appropriate to expand this areas to include some mention of the most reliable assays for accurate testosterone measurement (extraction and chromatography, followed by either MS or immunoassay)  

LL 82-83: This sentence is not the justification for the previous sentence. It mentions factors that may influence physiologic factors that influence circulating testosterone levels.  

LL 134-136 and LL 159-162: Suggest combining these two sections as they both deal with obesity and life style considerations.

LL 139-140: Would the author mention side effects and risks of using eflornithine? Also, please correct spelling.

LL149-153: This paragraph does not exactly recommend that metformin may be used to treat adolescent PCOS, although it seems to be implied. Must clarify whether metformin can be given alone or whether the presence of T2DM, IGT, or IR is necessary.

LL 155-156: Does this sentence refer to the effect of spironolactone alone or only in combination with metformin or COCPs?  

Author Response

Reviewer 1

This brief review of PCOS in adolescence addresses most of the issues regarding presentation, evaluation, and management of the disorder. It is unclear whether the brevity of the piece is due to space limitation or the choice of the author. Specific comments are as follows:

I can make this a more detailed review if the journal would like.  I was under the impression that there were word count limitations for the piece

LL 21-22: It would seem as if the discussion of criteria should be expanded to include, at least, Rotterdam criteria. This would raise the prevalence of PCOS to at least 20-24%. Also, Rotterdam criteria are mentioned later in the text.  

Many thanks for your comment. I have acknowledged that the Rotterdam criteria is the most commonly used and the other diagnostic criteria refer specifically to AYA (which is the focus of this review).  However I have expanded the reasons for this shift away in the same paragraph

‘The definition of polycystic ovarian morphology (PCOM) was 12 or more follicles on the ovary measuring 2-9mm, or an ovarian volume of at least 250px3.  It is now understood that 70% of the adolescent population would have PCOM based on these criteria(14). Clinical hyperandogenism is also a common feature of adolescence’..

LL33-35: This does not seem to be a complete sentence.

Many thanks. This has been amended

LL 67: “…hyperandrogenism…” is misspelled. Also, should the verb be “is” instead of “was”?

Many thanks. This has been amended

LL 76-78: Would the author include CAH, hyperthecosis, and exogenous androgen exposure?

This has been amended see below:

‘It is important for clinicians to exclude the presence of other pathologies, particularly those that may manifest with oligo-amenorrhoea and hyperandrogenism.  These include thyroid dysfunction, prolactinaemia, hypercortisolaemia, congenital adrenal hyperplasia (CAH), hyperthecosis, exogenous androgen exposure, or an androgen-producing tumour.  Clinical and biochemical investigations should differentiate these conditions.’

LL 81-82: It would appear appropriate to expand this areas to include some mention of the most reliable assays for accurate testosterone measurement (extraction and chromatography, followed by either MS or immunoassay)  

This has been included:

‘As a single measure calculated free testosterone, free androgen index (FAI), or calculated bioavailable testosterone are the most accurate tests in the assessment of biochemical hyperandrogenism in PCOS (29). Measurement of DHEAS and Androstenidione have a more limited role, but are useful in the exclusion of other causes of hyperandrogenism such a congenital adrenal hyperplasia (CAH), The most reliable assays include liquid chromatography-mass spectrometry (LCMS)/mass spectrometry and extraction/chromatography immunoassays (2).’

LL 82-83: This sentence is not the justification for the previous sentence. It mentions factors that may influence physiologic factors that influence circulating testosterone levels.  

Many thanks. This has been amended.

LL 134-136 and LL 159-162: Suggest combining these two sections as they both deal with obesity and life style considerations.

I have covered non pharmacological to pharmacological treatment so I have elected to keep this as per the original menuscript

LL 139-140: Would the author mention side effects and risks of using eflornithine? Also, please correct spelling.

Thank-you, the speling has been amended.  An additional sentence has been included:

‘However side effects include local skin irritation, pseudofolliculitis barbae, headache nausea, and occasionally urticaria and anaphylaxis.’

LL149-153: This paragraph does not exactly recommend that metformin may be used to treat adolescent PCOS, although it seems to be implied. Must clarify whether metformin can be given alone or whether the presence of T2DM, IGT, or IR is necessary.

Many thanks.  This has been amended to include the following:

Therefore metformin should be considered for AYA with PCOS symptoms including hirsuitism, metabolic derangement, and glucose intolerance. For symptoms of menstrual irregularity it is likely to be inferior to the COCP, but may improve symptoms compared to no treatment.

LL 155-156: Does this sentence refer to the effect of spironolactone alone or only in combination with metformin or COCPs?  

Many thanks. This has been clarified :

Spironolactone can be initiated as monotherapy in AYA with androgenic symptoms such as acne vulgaris (AV), or it can be used in combination with other topical and oral agents. The efficacy of spironolactone has been established by several studies showing improvement, with lesion reductions ranging from 50 to 100 percent in women treated for AV(51).

Reviewer 2 Report

The present review on PCOS in adolescence is a brief summary of what already published in this field without any new information or any innovative suggestion.

Some interesting and innovative arguments should be discussed such us the utility of the measurement of AMH, the utility of androstenedione, the utility of different cut-offs for testosterone in adolescence with respect to the adult age.

In addition, there are some inaccuracies and wrong messages, i.e. "acne is proposed as a signs of hyperandrogenism as hirsutism" or "metformin has been shown to be as effective as the COCP in the management of hirsutism" or "spironolactone has no effect on the metabolic dysregulation," that should be corrected.

The Author has to extensively review the manuscript.

Author Response

The present review on PCOS in adolescence is a brief summary of what already published in this field without any new information or any innovative suggestion.

Many thanks for your comments.  With respect this is a review article and the intention is not to introduce new findings in this field

Some interesting and innovative arguments should be discussed such us the utility of the measurement of AMH, the utility of androstenedione, the utility of different cut-offs for testosterone in adolescence with respect to the adult age.

Many thanks.  A paragraph has been included:

‘More recently Antimullerian hormone (AMH) has been used as a diagnostic tool in the assessment of PCOS.  Correlations between elevated AMH at age 16 with measures of testosterone, hirsuitism and menstrual irregularity, as well as diagnoses of PCOS at age 26 have been reported (38).  The utilisation of AMH as a test in adolescence needs to be considered carefully, with a wider range of normal seen in population based studies (39). ‘

Please see alterations in testosterone and androstenidione from Reviewer 1

In addition, there are some inaccuracies and wrong messages, i.e. "acne is proposed as a signs of hyperandrogenism as hirsutism" or "metformin has been shown to be as effective as the COCP in the management of hirsutism" or "spironolactone has no effect on the metabolic dysregulation," that should be corrected.

The Author has to extensively review the manuscript

This has been undertaken

Reviewer 3 Report

The author provides a timely review of PCOS in adolescence. Given the recent available consensus guidelines and growth of literature in this area, a review of this topic is appropriate at this time. Overall the manuscript is easy to read and does highlight some current important changes in the diagnosis and management of PCOS in this age group. My enthusiasm for the manuscript was dampened by the fact that the article felt to lack some level of detail throughout. Specific comments below:

Introduction, line 27: pathopneumonic should be pathognomic

Page 2, lines 45 - 49: The author should stress here that use of these diagnostic criteria are further complicated by the fact that normal adolescent ovarian morphology overlaps with that of PCOM making use of these criteria questionable. 

Page 2, lines 50-54: The author should make note of what these guidelines defined as "persistent."

Page 2, lines 55-59: The author should make note of recent published data that found the PES guidelines were best for identifying individuals with PCOS and metabolic syndrome. https://doi.org/10.1016/j.jpag.2019.01.006

Page 2, line 67: Hyperandrogenism is misspelled and use of the term "isolated mild hirsutism" would be more appropriate here.

Page 3, line 89: Timing of blood draw does not impact all hormone levels and the author should specify which levels are impacted by this.

Page 3, lines 94-95. Wording of this sentence is misleading and implies that all patients with PCOS should undergo ACTH stimulation test when only those with abnormal initial screening labs (17-OHP or clinical suspicion) should undergo this test.

Page 3, Co-morbidity: The author should really provide more detail about the type of metabolic and psychological sequlae described in this patient population. In addition, some recommendations for appropriate screening tools for psychological comorbidity could be added.

Page 3, management section, bullet (iii). This statement is incorrect. It is not recommended to defer diagnosis while offering symptom management, but rather deferring diagnosis while offering symptoms management is an "acceptable" alternative.

Author Response

Introduction, line 27: pathopneumonic should be pathognomic

Many thanks this has been amended

Page 2, lines 45 - 49: The author should stress here that use of these diagnostic criteria are further complicated by the fact that normal adolescent ovarian morphology overlaps with that of PCOM making use of these criteria questionable. 

Many thanks see changes below where this is expanded:

The definition of polycystic ovarian morphology (PCOM) was 12 or more follicles on the ovary measuring 2-9mm, or an ovarian volume of at least 250px3.  It is now understood that 70% of the adolescent population would have PCOM based on these criteria(14). Clinical hyperandogenism is also a common feature of adolescence. AND…

There has also been a revision in the ultrasound measurement of the follicle number per ovary (FNPO) with a recommendation of FNPO of at least 20 and / or ovarian volume of at least 10ml ensuring no dominant follicles are present (2).

Page 2, lines 50-54: The author should make note of what these guidelines defined as "persistent."

> 6 months.  This has been amended in the manuscript

Page 2, lines 55-59: The author should make note of recent published data that found the PES guidelines were best for identifying individuals with PCOS and metabolic syndrome. https://doi.org/10.1016/j.jpag.2019.01.006

Many thanks for your suggestion.  This is a retrospective review of 37 patients.  With respect, I cannot cite it as best evidence to support PES being the best criteria for identifying adolescents with PCOS

Page 2, line 67: Hyperandrogenism is misspelled and use of the term "isolated mild hirsutism" would be more appropriate here.

Many thanks the spelling error has been amended. I refer to hyperandrogenism as acne and hirsuitism.  Not hirsuitism alone. 

Page 3, line 89: Timing of blood draw does not impact all hormone levels and the author should specify which levels are impacted by this.

With respect, this is beyond the scope of this review article.  It is also a difficult undertake clinically as these young women often have menstrual irregularity and therefore it is difficult to time the blood draws.  I have made reference to the best test as a single measurement to make this the most clinically relevant to the readers.

Page 3, lines 94-95. Wording of this sentence is misleading and implies that all patients with PCOS should undergo ACTH stimulation test when only those with abnormal initial screening labs (17-OHP or clinical suspicion) should undergo this test.

Many thanks.  This has been amended to make this more clear to readers

Page 3, Co-morbidity: The author should really provide more detail about the type of metabolic and psychological sequlae described in this patient population. In addition, some recommendations for appropriate screening tools for psychological comorbidity could be added.

Many thanks, this has been amended.  Please see

comorbidity section:

‘Depressive and anxiety symptoms are more common in PCOS women compared to BMI matched subjects (44, 45) although these data are not focused on a adolescent population. It is uncertain whether the increased prevalence of these symptoms relate to acne, hirsuitism, increased BMI and infertility, which are linked to mood disturbance and emotional distress independently (2).  Or whether it is the association of the chronic nature of PCOS (46). In addition negative body image and eating disorders are more prevalent in adolescent PCOS, and clinicians should be cognisant of this when giving advice to AYA regarding lifestyle and weight loss (47).’  

and management section:

Particular tools for assessment of the psychological distress in AYA include:

·       Screening for Depression in Children and Adolescents: US Preventative Services Task Force Recommendations(60)

·       Common mental health problems: identification and pathways to care, NICE, 2011 (61)

·         Screening for and Treatment of Suicide Risk Relevant to Primary Care: A Systematic Review for the US Preventative Services Task Force (62)

Page 3, management section, bullet (iii). This statement is incorrect. It is not recommended to defer diagnosis while offering symptom management, but rather deferring diagnosis while offering symptoms management is an "acceptable" alternative.

This bullet point has been amended

Reviewer 4 Report

This review PCOS in adolescents by Dr Rebecca Deans, highlight a very important and clinically revelant topic, surrounded by controversy.  

My suggestion for manuscript improvement:

1- Revision of the english grammar and scientific language, there are misspelling words throughout the manuscript. For example hyperandrogenemia. Also the author use hyperandrogenism and hyperandrogenemia as an equivalent and they are clearly not

3- The section about AMH and PCOS is very confuse, should be rewrite

4- It is not clear the main objective of this paper as the author failed to highlight why is important to diagnosed PCOS during adolescent

Author Response

1- Revision of the english grammar and scientific language, there are misspelling words throughout the manuscript. For example hyperandrogenemia. Also the author use hyperandrogenism and hyperandrogenemia as an equivalent and they are clearly not

Many thanks, this has been reviewed and changed in the document.

3- The section about AMH and PCOS is very confuse, should be rewrite

Many thanks this has been re-written:

More recently Antimullerian hormone (AMH) has been studied in AYA as a predictor of adult PCOS.  Correlations have been made between an elevated AMH level at age 16, with measures of testosterone, clinical hirsuitism, menstrual irregularity, as well as diagnosis of PCOS at age 26 (38).  Although AMH shows promise as an adjunct to the diagnostic tests recommended for PCOS, the utilisation in adolescence needs to be considered carefully, as there is a wider range of normal values in AMH seen in the AYA, on population based studies (39). 

4- It is not clear the main objective of this paper as the author failed to highlight why is important to diagnosed PCOS during adolescent

Many thanks this is addressed in introduction:

Adult patients with PCOS commonly present with pathognomonic symptoms during adolescence. There is value in early identification, of PCOS to optimally manage the associated long-term metabolic and reproductive health sequalae (6).

Round 2

Reviewer 2 Report

The quality of the review improved. However, I still think that it is not innovative.

Author Response

The quality of the review improved. However, I still think that it is not innovative.

I have attempted to address the quality of this review.  Please see revised manuscript.